# Beyond Expectations: Recent Discovery of New Cave-Restricted Species Elevates the Água Clara Cave System to the Richest Hotspot of Subterranean Biodiversity in the Neotropics

**Rodrigo Lopes Ferreira** [1,2,*] and **Marconi Souza-Silva** [1,2]

1   Centro de Estudos em Biologia Subterrânea, Departamento de Ecologia e Conservação, Instituto de Ciências Naturais, Universidade Federal de Lavras, Campus Universitário, P.O. Box 3037, Lavras CEP 37200-000, MG, Brazil; marconisilva@ufla.br

2   Programa de Pós-Graduação em Ecologia Aplicada, Universidade Federal de Lavras, Lavras CEP 37200-000, MG, Brazil

\*   Correspondence: drops@ufla.br

**Abstract:** The Água Clara Cave System was previously recognized as a prominent hotspot of subterranean biodiversity in South America, harboring 31 cave-restricted species. However, a recent expedition conducted in September 2023, coinciding with an exceptionally dry period in the region, provided access to previously unexplored areas. Therefore, the objective of this research was to investigate the cave-restricted invertebrate species, extending the findings from a previous article on the Agua Clara Cave System published in June 2023, and emphasizing the significance of this system as one of the most crucial tropical biodiversity hotspots. This survey unveiled an additional 10 species, raising the count of cave-restricted species within the system to an impressive 41. This remarkable diversity not only solidifies the Água Clara Cave System's position as a paramount hotspot of subterranean biodiversity in the tropics but also serves as a stark warning about the imminent risks faced by these species. The escalating human-induced alterations in the region, notably deforestation, pose a significant risk to the survival of many of these unique and endemic species.

**Keywords:** obligate cave fauna; conservation; threats; species richness; stygobiont; troglobiont

The Água Clara Cave System (ACCS), situated in northeastern Brazil, has gained recognition as a "Hotspot of Subterranean Biodiversity" (HSB) due to its remarkable richness of cave-restricted fauna [1,2]. This cave system comprises a network of extensive subterranean conduits interconnected through one of the numerous fluvial outlets found on the eastern periphery of the Serra do Ramalho karst area in the western Bahia state. The system encopasses four caves: Gruna da Água Clara (13,880 m), Gruna dos Índios (510 m), Lapa dos Peixes I (9320 m), and Lapa dos Peixes II (2100 m) [1]. It is noteworthy that the cumulative length of all caves within the Agua Clara Cave System (ACCS) stands at 25,810 m. However, when accounting for the inaccessible and unexplored areas existing between the caves, it is plausible that the system might extend to more than 27 km.

The concept of HSB was originally introduced by Culver and Sket [3] to designate subterranean environments that host a minimum of 20 or more troglobitic/stygobitic species. More recently, there has been a movement to raise this threshold to a minimum of 25 cave-restricted species. However, this change has faced some criticism due to concerns about the arbitrary nature of the cutoff and the necessity to consider various parameters when defining subterranean hotspots [1]. These parameters encompass the need to consider different scales when identifying hotspots, the cave's latitudinal location, the lithology associated with the cave, the level of threat to subterranean habitats, and the number of cave-restricted species exclusive to that cave (or system) relative to the total number of troglobitic/stygobitic species it supports [1].

Most of the HSBs are situated in temperate regions, with only a smaller portion located in sub-tropical or tropical areas [4–11]. Additionally, it is noteworthy that most HSBs within these sub-tropical and tropical regions have only recently come to the forefront, signifying an upsurge in cave fauna research in these regions. For instance, the discovery of the first hotspots of subterranean biodiversity (HSB) in South America occurred only in recent years, following extensive sampling in the Areias cave system located in the southern São Paulo state (southeastern Brazil) and the Toca do Gonçalo cave in the northern Bahia state (northeastern Brazil) [11]. Presently, four recognized HSBs exist in South America: the Toca do Gonçalo cave, housing 22 cave-restricted species [11]; the Areias Cave System, hosting 31 cave-restricted species [9,11]; the ACCS, with a previously documented 31 cave-restricted species [2,11]; and the Igatu caves, harboring 37 cave-restricted species. It is important to note that the Igatu caves do not constitute a conventional, functionally interconnected "system" in the traditional sense. Instead, they comprise a collection of caves situated in a relatively small geographical area. Nevertheless, the significant number of cave-restricted species inhabiting these caves underscores their importance and emphasizes the urgency of conservation efforts [9].

Prior investigations conducted at the ACCS have identified a total of 31 cave-restricted species, spanning multiple taxonomic groups, including Hexapoda (9 species), Arachnida (7 species), Crustacea (6 species), Myriapoda (5 species), Gastropoda (2 species), Turbellaria (1 species), and Siluriformes (1 species) [1]. These species predominantly occupy terrestrial (22 species) habitats, with a smaller representation of semiaquatic (5 species) and aquatic species (3 species) [1]. Notably, only a fraction of this assemblage (11 species) has been formally described [12–22]. A recent expedition conducted in September 2023 coincided with an exceptionally dry period in the region, allowing access to previously unexplored sections of the system. This survey unveiled an additional 10 new species, increasing the tally of cave-restricted species within the system to an impressive 41 (Table 1). This number makes the ACCS the cave system with the highest richness of cave-restricted species in the Neotropical region. These newly documented species were encountered in Lapa dos Peixes I cave, within a conduit featuring a permanent water body (Figure 1A). The extreme drought conditions likely prompted many previously undocumented species to relocate from their original microhabitats, congregating near this subterranean "oasis", where they became observable. It is worth highlighting that this small, moisture-rich area of the cave hosted 24 cave-restricted species, which were found in a single day, using direct intuitive research. This observation suggests that during periods of severe external drought, this cave section plays a pivotal role in safeguarding several of the cave-restricted species found in the system. Furthermore, within this limited space, a substantial root mat system is present, serving as a consistent organic resource for the cave invertebrates (Figure 1B).

**Table 1.** Taxonomic diversity and distribution of 41 obligate cave species within the Águas Claras cave system, located in the northeastern region of Bahia state, Brazil (the newly recorded taxa in this study are highlighted in bold). The study includes observations from four specific caves within the cave system: Águas Claras cave (AC), Índios cave (IN), and Lapa dos Peixes (LP). Additionally, the study distinguishes fauna between terrestrial (T) and aquatic (A) or both habitats.

| Phylum/Classis | Order | Family | Species and Morphotypes | AC | IN | LP II | LP I | Habitat |
|---|---|---|---|---|---|---|---|---|
| Platyhelminthes | Tricladida | Dugesiidae | *Girardia spelaea* | | | + | + | A |
| Arachnida | Acari | Rhagidiidae | Rhagidiidae sp.1 | + | | | + | T |
| | Amblypygi | Charinidae | *Charinus troglobius* | + | | | + | T |
| | Araneae | Caponiidae | Caponiidae sp.1 | + | | | + | T |
| | | Ochyroceratidade | Ochyroceratidae sp.1 | + | | + | + | T |
| | | Ochyroceratidade | **Ochyroceratidae sp.2** | | | | + | T |
| | | Palpimanidae | **Palpimanidae sp.1** | | | | + | T |
| | | Tetrablemmidae | **Tetrablemmidae sp.1** | | | | + | T |
| | Opiliones | Gonyleptidae | *Giuponnia chagasi* | + | | + | | T |
| | Palpigradi | Eukoeneniidae | *Eukoenenia* sp.1 | + | | + | + | T |
| | Pseudoscorpiones | Ctoniidae | *Pseudochthonius koinopoliteia* | + | | | + | T |

**Table 1.** *Cont.*

| Phylum/Classis | Order | Family | Species and Morphotypes | AC | IN | LP II | LP I | Habitat |
|---|---|---|---|---|---|---|---|---|
| Collembola | Symphypleona | Sminthuridae | *Troglobentosminthurus luridus* | + | | + | | T |
| | | Sminthuridae | Sminthuridae sp.1 | | | | + | T |
| | | unidentified | Symphypleona sp.1 | + | | | | T |
| | Entomobryomorpha | | Entomobryomorpha sp.1 | + | | | + | T |
| | | | Entomobryomorpha sp.2 | + | + | | | T |
| Diplura | | Projapygidae | **Projapygidae sp.1** | | | | + | T |
| Insecta | Blattodea | unidentified | Blattodea sp.1 | + | | + | + | T |
| | Coleoptera | Carabidae | **Clivinina sp.1** | | + | | + | T |
| | Coleoptera | Carabidae | **Trechinae sp.1** | | | | + | T |
| | Dermaptera | Diplatyidae | *Mesodiplatys falcifer* | | | | + | T |
| | Ensifera | Phalangopsidae | *Endecous infernalis* | + | + | + | + | T |
| | Hemiptera | Delphacidae | **Delphacidae sp.1** | | | | + | T |
| | Hemiptera | Hydrometridae | ***Spelaeometra* sp.1** | | | | + | T |
| | Hymenoptera | Formicidae | *Nylanderia* sp.1 | + | | + | + | T |
| Crustacea | Isopoda | Styloniscidae | *Pectenoniscus carinhanhensis* | + | + | | + | T |
| | | Styloniscidae | Styloniscidae sp.1 | + | | | + | T/A |
| | | Styloniscidae | Styloniscidae sp.2 | + | | + | + | T/A |
| | | Styloniscidae | Styloniscidae sp.3 | | | + | + | T/A |
| | | Styloniscidae | *Xangoniscus aganju* | + | | + | + | T/A |
| | | Plathyarthridae | *Trichorhina* sp.1 | + | + | | | T |
| Myriapoda | Geophilomorpha | unidentified | Geophilomorpha sp.1 | | | + | | T |
| | Polydesmida | Chelodesmidae | *Cayenniola* sp.1 | + | + | + | + | T |
| | | Trichopolidesmidae | *Phaneromerium* sp.1 | | + | + | + | T |
| | | Trichopolidesmidae | **Trichopolidesmidae sp.1** | | | | + | T |
| | | Pyrgodesmidae | Pyrgodesmidae sp.1 | + | | + | + | T |
| | | Oniscodesmidae | Oniscodesmidae sp.1 | | | | + | T |
| | Siphonophorida | Siphonophoridae | **Siphonophoridae sp.1** | | | | + | T |
| Mollusca | Gastropoda | Pomatiopsidae | *Spiripockia punctata* | | | + | + | A |
| | | unidentified | Eupulmonata sp.1 | + | | | + | A |
| Osteichthyes | Siluriformes | Trichomycteridae | *Trichomycterus rubbioli* | + | | | + | A |

The newly discovered species encompass a diverse array of taxa, all of which represent new taxa, with some already in the process of formal description. These include a palpimanid spider (Araneae: Palpimanidae—Figure 2A), an ochiroceratid spider (Araneae: Ochiroceratidae—Figure 2B), a tetrablemmid spider (Araneae: Tetrablemmidae: *Matta* sp.—Figure 2C), a siphonophorid millipede (Diplopoda: Siphonophorida: Siphonophoridae—Figure 2D), a trichopolydesmid millipede (Diplopoda: Polydesmida: Trichopolydesmidae), a delphacid planthopper (Hemiptera: Delphacidae—Figure 2E), two carabid beetles (Coleoptera: Carabidae: Clivinina—Figure 2F and Trechinae), a hydrometrid bug (Hemiptera: Hydrometridae: *Spelaeometra* sp.—Figure 2G), and a projapygid (Diplura: Projapygidae—Figure 2H).

All these species exhibited typical troglomorphic traits, including reduced or absent eyes and pigmentation. Additionally, specific troglomorphic characteristics were observed. The palpimanid spider displayed extremely reduced eyes and weak pigmentation, contrasting with the general morphology observed in the remaining species of this family. Similar traits were observed in the ochiroceratid and tetrablemmid spiders, which also exhibited an additional thickening of the cuticle and no eyes. Both the siphonophorid and the trichopolydesmid millipedes were completely unpigmented. They also showed an increase in sensory pits on the antennal segments and an unusually long tergal setae, considered troglomorphic traits in other millipede taxa [23–25]. The delphacid planthopper exhibited all the typical troglomorphic traits observed in other cave-restricted planthoppers, including the absence of eyes, pigment reduction, and wings reduction [26–30]. Both carabid beetles displayed typical troglomorphic traits, such as eye and pigment reduction, as well as wing reduction [31,32]. The hydrometrid bug (*Spelaeometra* sp.) exhibited all the troglomorphic traits observed in the other two known species of the genus. This included reduced eyes and pigmentation, along with elongated legs and antennae [33,34]. Finally,

the projapygid (Diplura) exhibited the most traditional troglomorphic traits among diplurans, such as appendage elongation and an increase in sensory setae at the antennal segments [35]. Unfortunately, the specimen lost both cerci when discovered above a rock on the cave floor (Figure 2H), but one cercus was later recovered, indicating considerable elongation.

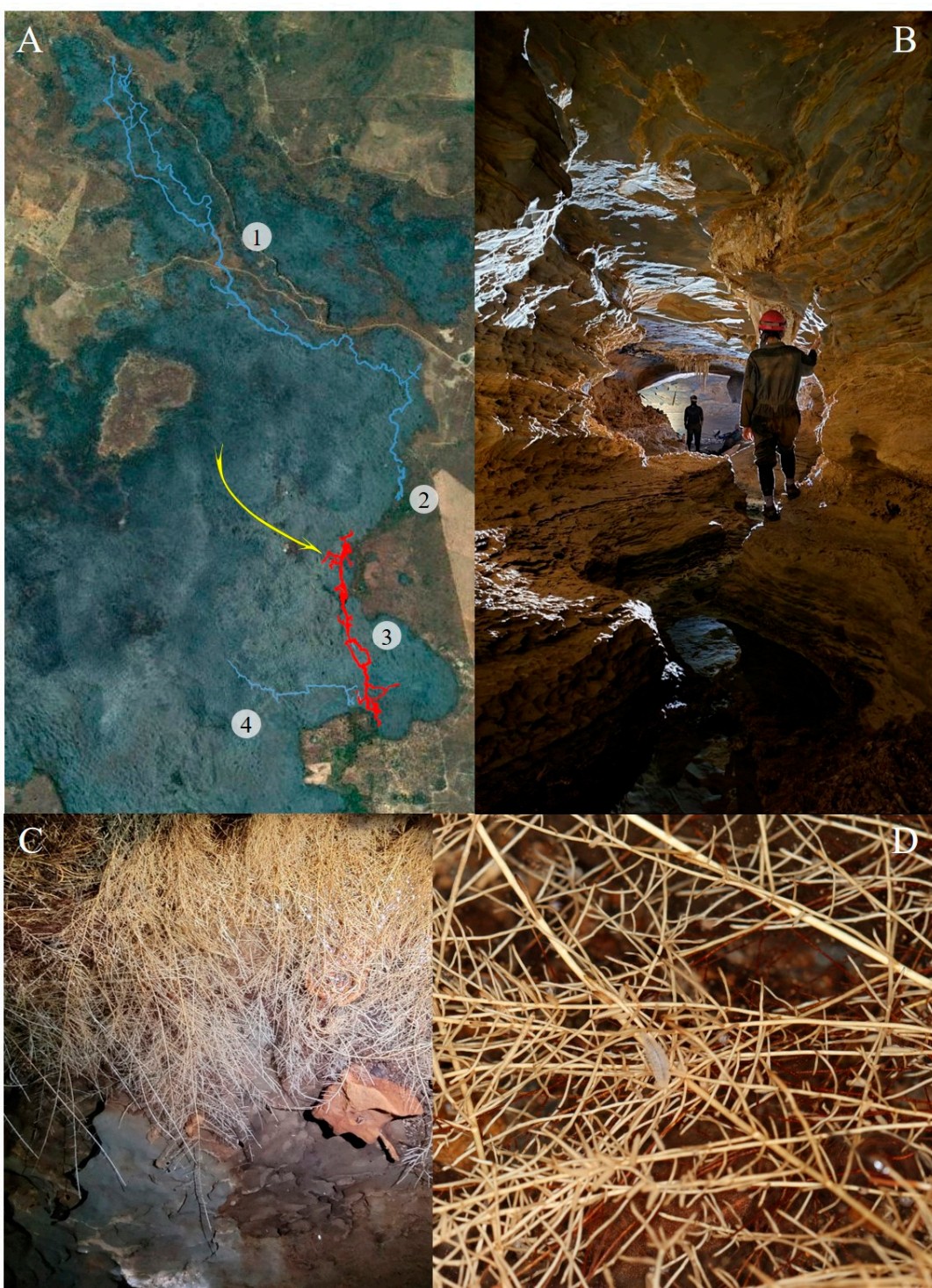

**Figure 1.** Água Clara cave system: (**A**) spatial distribution of the caves Gruna da Água Clara (1), Gruna dos Índios (2), Lapa dos Peixes I (3), and Lapa dos Peixes II (4); the yellow arrow indicates the region where the newly discovered cave-restricted species were found; (**B**) conduit where the newly discovered species were found (notice the water pond on the floor of the conduit); (**C**) root mats covering the cave floor; (**D**) a close-up view of a root mat with a troglobitic isopod (*Xangoniscus* sp.).

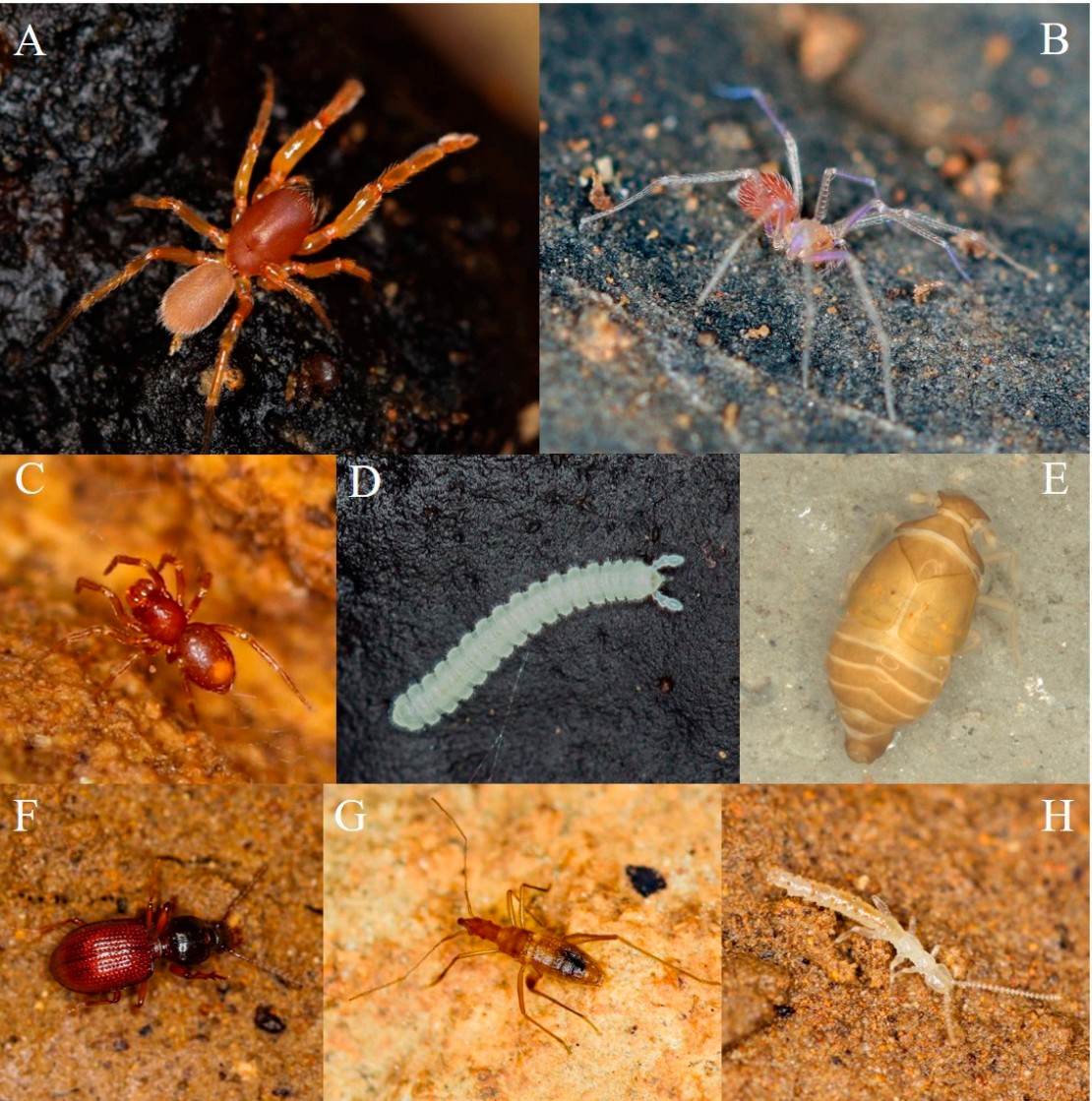

**Figure 2.** Newly discovered cave-restricted species from the ACCS: (**A**) Palpimanidae (Araneae); (**B**) Ochiroceratidae (Araneae); (**C**) Tetrablemmidae (Araneae); (**D**) Siphonophoridae (Diplopoda: Siphonophorida); (**E**) Delphacidae (Hemiptera); (**F**) Clivinina (Coleoptera: Carabidae); (**G**) *Spelaeometra* sp. (Hemiptera: Hydrometridae); (**H**) Projapygidae (Diplura).

While troglomorphisms can serve as valuable indicators of the potential status of these species, their analysis should always consider the contexts of the external ecosystems surrounding the caves. For instance, if a species, completely depigmented, blind, and with a reduced cuticle, is discovered in a cave within a humid forest (like the Amazon rainforest), these traits may not necessarily signify its restriction to that cave. This is because the surrounding forest provides numerous shaded and humid microhabitats (such as spaces under logs, leaf litter, etc.) that could easily accommodate individuals of this species. On the contrary, if a species with similar characteristics were found in a cave located in an arid or semi-arid region, these morphological traits would strongly indicate its restriction to subterranean habitats. This is because, in the surrounding epigean environments, such organisms would rarely encounter suitable microhabitats for their survival.

Thus, considering the highly xeric epigean environment surrounding the ACCS (Figure 1A), it is unlikely that these species (as well as the other 31 troglomorphic species previously registered in the system) can maintain viable populations on the surface. Therefore, not only were troglomorphic traits instrumental in identifying these species as troglo-

bitic, but also the external surrounding conditions, which are highly restrictive, imposing physiological constraints and preventing the occurrence of these species in external habitats. Finally, it is noteworthy that some of the newly discovered species were examined by taxonomists who confirmed their status (as mentioned in the Acknowledgements section).

It is important to note that some of these newly discovered species hold particular significance, such as the palpimanid spider, which marks the first known troglobitic species within this family worldwide. Additionally, the presence of the delphacid planthopper in this region is noteworthy, as the three previously documented subterranean-restricted species from this family were exclusively recorded in New Caledonia [26,27].

Therefore, the Gruna da Água Clara cave, once recognized for harboring the highest troglobitic species richness among ACCS caves, with a total of 23 species, has now been surpassed by the Lapa dos Peixes I cave, which hosts 35 species. The Lapa dos Peixes II cave accommodates 17 species, whereas only 5 species have been found thus far in the Gruna dos Índios cave.

The recent discovery of ten additional species within the ACCS raises a red flag on two critical fronts. Firstly, it underscores the extraordinary diversity of this system, currently facing severe threats driven by an unprecedented increase in various anthropogenic impacts, particularly in recent decades [1]. Urgent actions, such as the establishment of a fully protected conservation unit, are imperative. Secondly, and perhaps more significantly, this discovery serves as a powerful reminder that limited samplings are insufficient when attempting to unveil the true extent of species richness within a cave or cave system. In Brazil, caves are increasingly at risk due to various industrial activities, such as mining and hydropower dam construction, among others. To assess which caves should be safeguarded, a mere two samplings are currently required to determine their relevance. Consequently, in Brazil, a cave may be deemed of low significance simply because of an inadequate sampling effort. Therefore, it is crucial to demand additional samplings during the environmental licensing processes to more accurately assess the relevance of a cave, especially given that Brazilian caves have never faced such high levels of threat as they do today [36].

**Author Contributions:** Conceptualization, R.L.F.; data acquisition, R.L.F. and M.S.-S.; original draft preparation, R.L.F.; review and editing, R.L.F. and M.S.-S. All authors have read and agreed to the published version of the manuscript.

**Funding:** The authors would like to thank the Centro Nacional de Pesquisa e Conservação de Cavernas (CECAV) and Instituto Brasileiro de Desenvolvimento e Sustentabilidade (IABS) for their financial support (TCCE ICMBio/Vale 01/2018). We are also thankful to the CNPq (National Council for Scientific and Technological Development, grant no. 302925/2022-8) for the productivity scholarship provided to R.L.F., and to the team from the Center of Studies in Subterranean Biology (CEBS/UFLA) for their support in the field trips.

**Institutional Review Board Statement:** Not applicable.

**Informed Consent Statement:** Not applicable.

**Data Availability Statement:** No new data were created or analyzed in this manuscript. Data sharing is not applicable for this manuscript.

**Acknowledgments:** The authors would like to thank the team from the Center of Studies in Subterranean Biology (CEBS/UFLA) for their support in the field trips. We also thank Gabriel Augusto Silva Vaz, Paulo César Reis Venâncio, and Priscila Emanuela Souza for their cooperation in field surveys. Finally, we would like to extend our sincere gratitude to the taxonomists who confirmed the cave-restricted status for some of the newly discovered species and who are currently describing some of them, including Leonardo Sousa Carvalho (Araneae), Leticia Vieira (Coleoptera), and Júlio César Do Carmo Vaz Santos (Auchenorrhynca).

**Conflicts of Interest:** The authors declare no conflict of interest.

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
