# Peer review of "Beyond Expectations: Recent Discovery of New Cave-Restricted Species Elevates the Água Clara Cave System to the Richest Hotspot of Subterranean Biodiversity in the Neotropics"

_diversity, doi:10.3390/d15121215_

Round 1

Reviewer 1 Report (Previous Reviewer 3)

Comments and Suggestions for Authors

The authors took into account the reviewer's comments and improved the manuscript. The added paragraphs and the table also contributes to the informativeness of the article.

Regarding the table:

1) The first column is labeled as "Taxons," and the second as "Taxon." What's the difference? This should be either fixed or explained. For instance, the first column could be labeled as "Phylum and Classis" and the second column as "Order."

2) Why is the taxon Collembola in column 1 placed separately from Hexapoda? This is incorrect; they belong to Hexapoda as an independent class.

3) Why is the numbering of some unidentified taxa (sp.) not in order? For instance, "Entomobryomorpha sp.4" and "sp.7" are included, but where are "Entomobryomorpha sp.1-3" and "sp.5-6"? This may need clarification in the table captions or fixed. Also please check other similar cases. 

Author Response

Dear Reviewer,

I want to express my sincere appreciation for your invaluable feedback on the manuscript. In this resubmitted version, we thoroughly addressed all the suggestions you provided.

As recommended, we replaced "taxons" with "Phylum and Classis" in the first column and "Order" in the second column. Additionally, we rectified the taxa in the table, retaining "Collembola" as a Class, excluding "Diplura" from "Hexapoda," treating it as a separate Class, and substituting "Hexapoda" with "Insecta" to specifically denote insects. Finally, we renumbered the morphospecies so that the numbering is now sequential when referring to the same taxon (as a family).

We trust that the manuscript is now in an acceptable form for publication in the Journal of Diversity.

Reviewer 2 Report (Previous Reviewer 2)

Comments and Suggestions for Authors

Dear Authors,

with the changes made in some parts of the text and the inclusion of the table of species and their distribution, it is my opinion that the manuscript now contains all the necessary information and can be published.

Thanks for your work

Kind regards

Author Response

Dear Reviewer,

I extend my gratitude for your invaluable feedback on the manuscript. In this resubmitted version, we diligently addressed all the suggestions you provided. We believe that the manuscript is now in an acceptable form for publication in the Journal of Diversity.

Best regards,

Rodrigo

Reviewer 3 Report (Previous Reviewer 1)

Comments and Suggestions for Authors

I do appreciate the improvement of the manuscript. Thank you for that.

I.                I am still not very happy with the term semiaquatic/amphibious. These isopods (T/A in Table 1) seem rather terrestrial to me. I had looked in publications, but I could not really figure out why are they amphibious. To me, this term refers to animals (such as amphibians) that can live long periods of time under the water, breathing through their tegument or their lungs. My question is: do these isopods swim under the water for longer periods? Are they present on water surface like the springtails? Do they dive very shortly for preying on microorganisms?

In any case, I am sure you, the authors, have observed the behavior in situ and you may easily say whether they are really amphibious. If they are present on water surface, or if they only dive shortly, they are not amphibious. If they stay longer under the water surface, then they are amphibious indeed, and you can use this term back in the manuscript (so ‘amphibious’ is fine. Use this term as it is present in other publications).

Here, in the 4th row, add an ‘s’ (Styloniscidae p.4). So, ‘sp.4’

II.              Introducing Table 1 is the largest improvement of the manuscript, even if it was provided also in June. An even larger improvement would be adding of an additional column that will tell whether the animal is troglo/stygobiontic or troglo/stygophilic. This column will be helpful.

Author Response

Dear Reviewer,

I extend my gratitude for your invaluable feedback on the manuscript. In this resubmitted version, we diligently addressed all the suggestions you provided. We believe that the manuscript is now in an acceptable form for publication in the Journal of Diversity.

Regarding your concerns about the term "semiaquatic/amphibious" in reference to the isopods: in Brazilian caves, the majority of troglobitic species belonging to the family Styloniscidae are indeed "amphibious." This term has been consistently used by taxonomists over the last few years when describing species from Brazil. It signifies that while these species are capable of dispersing through terrestrial habitats, they are predominantly found submerged in aquatic environments. These habitats range from cave streams (e.g., Xangoniscus aganju, Xangoniscus lundi, Xangoniscus santinhoi, and Xangoniscus odara) to phreatic waters such as cave "ponds" (e.g., Chaimowiczia uai, Iuiuniscus iuiuensis) and travertine pools (e.g., Xangoniscus itacarambiensis, Xangoniscus dagua and Xangoniscus ibiracatuensis). The only genuinely terrestrial cave-restricted styloniscid genera known in Brazilian caves to date are Pectenoniscus and Cylindroniscus.

Therefore, all Styloniscidae observed in the ACCS, except for the species Pectenoniscus carinhanhensis, are typically observed underwater, either in ponds or travertine pools. As mentioned earlier, these isopods can disperse through terrestrial moistened substrates within the cave if the water body where they primarily dwell undergoes drying, as they seek alternative ponds. However, it is crucial to emphasize that such dispersive movements were rarely observed in the ACCS, even with the extensive sampling expeditions conducted in this cave system over the last two decades.

Below, you will find references describing various species of such "amphibious" cave styloniscids from Brazil, accompanied by photographs of their respective habitats and detailed descriptions. We do not foresee any issues with using the term "semi-aquatic" as suggested, and that is why we made the change as per your previous recommendation. Nevertheless, it is important to note that the most commonly used term in literature when referring to such species is, in fact, "amphibious," as demonstrated in the references below.

Regarding your suggestion to include an extra column in table 1 indicating whether the species is troglo/stygobiontic or troglo/stygophilic: all species listed in the table are restricted to caves, making them either troglobitic or stygobitic. As the primary habitat (terrestrial/aquatic) is already specified, we believe it is unnecessary to add an additional column for this distinction.

Once again, thank you for your thorough review, and we look forward to any further guidance you might provide.

Best regards,

Rodrigo

Souza, L. A., Ferreira, R. L., & Senna, A. R. (2015). Amphibious shelter-builder Oniscidea species from the New World with description of a new subfamily, a new genus and a new species from Brazilian Cave (Isopoda, Synocheta, Styloniscidae). PloS one, 10(5), e0115021.

Bastos-Pereira, R., Souza, L. A., & Ferreira, R. L. (2017). A new amphibious troglobitic styloniscid from Brazil (Isopoda, Oniscidea, Synocheta). Zootaxa4294 (2): 292-300.

Silva, A. P. B., Oliveira, I. P. M. R., Bastos-Pereira, R., & Ferreira, R. L. (2018). Are laboratory studies on behavior of troglobitic species always trustful? A case study with an isopod from Brazil. Behavioural processes, 153, 55-65.

Campos-Filho, I. S., Fernandes, C. S., Cardoso, G. M., Bichuette, M. E., Aguiar, J. O., & Taiti, S. (2019). Two new species and new records of terrestrial isopods (Crustacea, Isopoda, Oniscidea) from Brazilian caves. Zootaxa, 4564(2), zootaxa-4564.

Cardoso, G. M., Bastos-Pereira, R., Souza, L. A., & Ferreira, R. L. (2020). New troglobitic species of Xangoniscus (Isopoda: Styloniscidae) from Brazil, with notes on their habitats and threats. Zootaxa, 4819(1), zootaxa-4819.

Campos-Filho, I. S., Gallo, J. S., Gallão, J. E., Torres, D. F., Horta, L., Carpio-Díaz, Y. M., ... & Bichuette, M. E. (2022). Unique and fragile diversity emerges from Brazilian caves–two new amphibious species of Xangoniscus Campos-Filho, Araujo & Taiti, 2014 (Oniscidea, Styloniscidae) from Serra do Ramalho karst area, state of Bahia, Brazil. Subterranean Biology, 42, 1-22.

This manuscript is a resubmission of an earlier submission. The following is a list of the peer review reports and author responses from that submission.

Round 1

Reviewer 1 Report

Comments and Suggestions for Authors

Several observations:

In this article, the authors add ten species to the diversity list of Agua Clara Cave System (ACCS). Recommendations are made in order to protect this tropical biodiversity hotspot.

Title: Use ‘hotspot’ to plural here. So, ‘hotspots’

“… to one of the richest hotspots of subterranean biodiversity”

Abstract: Add at the end of the abstract a sentence stating clearly the aim. Maybe something like “The aim of this research was to inventorize the invertebrate species that depend on this cave environment and to draw attention on the importance of the Agua Clara Cave System, as one of the world most important biodiversity hotspots”.

Explain already in the abstract that this article is a continuation of the previous article published in June 2023.

Page 2, first paragraph:

Make this sentence (Two additional areas also warrant mention, albeit representing unique circumstances) clearer:

Two additional areas are also worth mentioning, albeit representing unique circumstances.

Page 2, second paragraph:

Remove ‘amphibious’

What animals are considered amphibious? would stick to terrestrial and aquatic, as to me 'amphibious' tell more on frogs.

You can consider adding the species list again in this article with the 10 new species included. Indicate which are the 10 new species, which species are new to science, which species are endemic, troglobitic/stygobitic etc.

Remove funding information from Acknowledgements. There’s a separate section on Funding.

Comments on the Quality of English Language

English is OK.

Author Response

Dear reviewer,

We are submitting a revised draft of our manuscript entitled, “Beyond expectations: recent discovery of new cave-restricted species elevates the Água Clara Cave System to one of the richest hotspot of subterranean biodiversity in the tropics”. We appreciate your time and thorough review of the manuscript. It is essential to mention that all grammatical corrections have been directly incorporated into the text and are indicated by 'track changes.' Therefore, we have chosen not to list each one below. Instead, we have addressed your specific questions and provided responses to your general comments.

Title: Use ‘hotspot’ to plural here. So, ‘hotspots’… to one of the richest hotspots of subterranean biodiversity”

Response: Changed as suggested.

 Abstract: Add at the end of the abstract a sentence stating clearly the aim. Maybe something like “The aim of this research was to inventorize the invertebrate species that depend on this cave environment and to draw attention on the importance of the Agua Clara Cave System, as one of the world most important biodiversity hotspots”. Explain already in the abstract that this article is a continuation of the previous article published in June 2023.

 Response: We have incorporated to the abstract the following text: “Therefore, the objective of this research was to inventory the cave-restricted invertebrate species, extending the findings from a previous article on the Agua Clara Cave System published in June 2023, and emphasizing the significance of this system as one of the most crucial tropical biodi-versity hotspots.”

Page 2, first paragraph:

Make this sentence (Two additional areas also warrant mention, albeit representing unique circumstances) clearer:Two additional areas are also worth mentioning, albeit representing unique circumstances.

Response: Changed as suggested.

Page 2, second paragraph:

Remove ‘amphibious’. What animals are considered amphibious? would stick to terrestrial and aquatic, as to me 'amphibious' tell more on frogs.

Response: The term ‘amphibious’ was replaced by “semiaquatic”.

You can consider adding the species list again in this article with the 10 new species included. Indicate which are the 10 new species, which species are new to science, which species are endemic, troglobitic/stygobitic etc.

Response: The table was inserted in the manuscript, as sugested.

Remove funding information from Acknowledgements. There’s a separate section on Funding.

Response: Changed as suggested.

Reviewer 2 Report

Comments and Suggestions for Authors

As the manuscript was prepared for a special issue on subterranean biodiversity hotposts, it certainly increases previous knowledge by reporting interesting data. I therefore recommend its publication.

Some suggestions for improvement:

1) Include a table with a complete list of new and old found troglo-stygobiont species and an indication in which cave of the complex they were found.

2) Even if already published in the previous paper, insert a figure with the location of the site and a sketch of the caves.

3) Include between the citations of the tropical caves considered HSB (from line 47 onwards) also the new caves recently published in the same issue (Hotspots of Subterranean Biodiversity II)

Best wishes

Author Response

Dear reviewer,

We are submitting a revised draft of our manuscript entitled, “Beyond expectations: recent discovery of new cave-restricted species elevates the Água Clara Cave System to one of the richest hotspot of subterranean biodiversity in the tropics”. We appreciate your time and thorough review of the manuscript. It is essential to mention that all grammatical corrections have been directly incorporated into the text and are indicated by 'track changes.' Therefore, we have chosen not to list each one below. Instead, we have addressed your specific questions and provided responses to your general comments.

1) Include a table with a complete list of new and old found troglo-stygobiont species and an indication in which cave of the complex they were found.

Response: Included as suggested.

2) Even if already published in the previous paper, insert a figure with the location of the site and a sketch of the caves.

Response: Inserted as suggested.

3) Include between the citations of the tropical caves considered HSB (from line 47 onwards) also the new caves recently published in the same issue (Hotspots of Subterranean Biodiversity II)

Response: We included all the tropical caves considered HSB, as suggested.

Reviewer 3 Report

Comments and Suggestions for Authors

The paper is an interesting addition to the previously published paper (https://doi.org/10.3390/d15060761) aiming to indicate one of the most diverse cave hotspots in the tropics. It is well written and illustrated.

Some suggestions to improve:

1) When describing the cave system, please provide the total length and vertical amplitude (depth) if available.

2) Please, specify the sampling methods used.

3) Please, specify whether the newly discovered species have undergone preliminary taxonomic study. Are they new species? Can they be identified to the genus level?

These and some other remarks are noted in the PDF of the reviewed manuscript.

Author Response

We are submitting a revised draft of our manuscript entitled, “Beyond expectations: recent discovery of new cave-restricted species elevates the Água Clara Cave System to one of the richest hotspot of subterranean biodiversity in the tropics”. We appreciate your time and thorough review of the manuscript. It is essential to mention that all grammatical corrections have been directly incorporated into the text and are indicated by 'track changes.' Therefore, we have chosen not to list each one below. Instead, we have addressed your specific questions and provided responses to your general comments.

Some suggestions to improve:

1) When describing the cave system, please provide the total length and vertical amplitude (depth) if available.

Response: We included the total length of the system, but the vertical amplitude (depth) is not available unfortunately.2) Please, specify the sampling methods used.

3) Please, specify whether the newly discovered species have undergone preliminary taxonomic study. Are they new species? Can they be identified to the genus level?

Response: All new species discovered are new to science. Most were already sent to specialists and are under description (those with specimens suitable for description – the palpimanid spider, for instance, was a female, so we will try to find a male for describing the species). Genus was only determined now, for the Spelaeometra species.

These and some other remarks are noted in the PDF of the reviewed manuscript.

Response: Thank you! All grammatical corrections have been directly incorporated into the text and are indicated by 'track changes.
